# Xanthine Oxidase Inhibitory Peptides from *Larimichthys polyactis*: Characterization and In Vitro/In Silico Evidence

**DOI:** 10.3390/foods12050982

**Published:** 2023-02-25

**Authors:** Xiaoling Chen, Weiliang Guan, Yujin Li, Jinjie Zhang, Luyun Cai

**Affiliations:** 1College of Food and Pharmaceutical Sciences, Ningbo University, Ningbo 315211, China; 2Ningbo Innovation Center, College of Biosystems and Food Science, Zhejiang University, Ningbo 315100, China; 3College of Biological and Chemical Engineering, Zhejiang Engineering Research Center for Intelligent Marine Ranch Equipment, NingboTech University, Ningbo 315100, China; 4College of Food Science and Engineering, Ocean University of China, Qingdao 266003, China

**Keywords:** xanthine oxidase inhibitor, peptides, small yellow croaker, identification, molecular docking

## Abstract

Hyperuricemia is linked to a variety of disorders that can have serious consequences for human health. Peptides that inhibit xanthine oxidase (XO) are expected to be a safe and effective functional ingredient for the treatment or relief of hyperuricemia. The goal of this study was to discover whether papain small yellow croaker hydrolysates (SYCHs) have potent xanthine oxidase inhibitory (XOI) activity. The results showed that compared to the XOI activity of SYCHs (IC_50_ = 33.40 ± 0.26 mg/mL), peptides with a molecular weight (MW) of less than 3 kDa (UF-3) after ultrafiltration (UF) had stronger XOI activity, which was reduced to IC_50_ = 25.87 ± 0.16 mg/mL (*p* < 0.05). Two peptides were identified from UF-3 using nano-high-performance liquid chromatography–tandem mass spectrometry. These two peptides were chemically synthesized and tested for XOI activity in vitro. Trp-Asp-Asp-Met-Glu-Lys-Ile-Trp (WDDMEKIW) (*p* < 0.05) had the stronger XOI activity (IC_50_ = 3.16 ± 0.03 mM). The XOI activity IC_50_ of the other peptide, Ala-Pro-Pro-Glu-Arg-Lys-Tyr-Ser-Val-Trp (APPERKYSVW), was 5.86 ± 0.02 mM. According to amino acid sequence results, the peptides contained at least 50% hydrophobic amino acids, which might be responsible for reducing xanthine oxidase (XO) catalytic activity. Furthermore, the inhibition of the peptides (WDDMEKIW and APPERKYSVW) against XO may depend on their binding to the XO active site. According to molecular docking, certain peptides made from small yellow croaker proteins were able to bind to the XO active site through hydrogen bonds and hydrophobic interactions. The results of this work illuminate SYCHs as a promising functional candidate for the prevention of hyperuricemia.

## 1. Introduction

Uric acid has been identified as a recognized or prospective biomarker for various pathological conditions. Lifestyle factors such as high fructose intake, alcohol addiction, and a high-purine diet can all contribute to high levels of uric acid [1]. Hyperuricemia develops when serum uric acid concentrations surpass solubility limits (6.8 mg/dL at physiological pH). Chronic hyperuricemia may raise the risk of gout, which can lead to gout stones, acute arthritis, and other complications [2,3]. The major pathway of uric acid regulation involves the modulation of purine metabolism via xanthine oxidase (XO, EC 1.17.3.2), which is a molybdenum-containing homodimeric cytoplasmic enzyme with a molecular weight (MW) of approximately 300 kDa [4,5]. It predominantly catalyzes the conversion of xanthine and hypoxanthine to uric acid in the human body. Therefore, substances effectively inhibiting XO can be used to prevent hyperuricemia, as exemplified by drugs such as allopurinol, which can provide short-term relief from the pain caused by gout [6]. However, these synthetic drugs often cause a variety of negative effects; for example, allopurinol is highly susceptible to drug cross-reactivity and may cause rashes [7,8].

As a consequence, researchers are trying to create new inhibitors from natural sources that are safe, effective, and less expensive, such as food-derived bioactive peptides with a high XO inhibitory (XOI) effect and minimal side effects. XOI peptides are generally derived from protein hydrolysates by separation, purification, and identification, and include dairy products [9], nuts [10], and aquatic products [11,12]. For example, the peptides YF, WPDARG, ACECD, and FPSV were discovered in the hydrolysates of aquatic products and have been shown to alleviate hyperuricemia [11,13,14]. These bioactive peptides generated from dietary protein hydrolysates are typically easily absorbed and are safer than pharmaceuticals [15,16]. Furthermore, quantitative structure–activity relationships and molecular docking approaches, which are widely used in the screening and discovery of natural small molecule active compounds, have contributed to the illumination of new peptides. Thus, it is important to explore physiologically active peptides from aquatic materials.

The small yellow croaker (*Larimichthys polyactis*, SYC), a Sciaenidae fish, is widely distributed as a benthic warm temperate fish in the coastal waters of China [17], favored by consumers because of its high nutritional value, umami taste, and tender texture [18]. However, certain characteristics of SYC limit its current utilization, such as its small size, susceptibility to perishability, and potent fishy smell [19]. Typically, it is processed into items such as fish cake, fish balls, and canned food [20,21,22]. Therefore, to improve the utilization and economic value of SYC, it is critical to develop higher-value-added products. 

In this work, we combined traditional testing methods with computer simulation techniques to acquire XOI peptides from SYC muscle. First, papain hydrolysates from SYC were graded by ultrafiltration (UF) technology. Next, the group with the greatest XOI effect was identified and the amino acid sequences of two peptides were obtained. The contribution of the synthesized peptides to the XOI activities of SYC was calculated. Furthermore, molecular docking analysis was used to model the interactions between these peptides and the XO active site and, thus, shed light on the XO inhibition mechanisms of SYC peptides. The XOI effects of the XOI peptides from SYC in vitro were elucidated.

## 2. Materials and Methods

### 2.1. Materials and Chemicals

Frozen SYC 9 ± 1 cm in length was obtained from the Zhejiang Xianghai Food Co., Ltd. in Wenzhou, China. Papain (100,000 U/g), Alcalase (100,000 U/g), Neutrase (50,000 U/g), and xanthine oxidase (X1875-5UN, derived from bovine milk) were purchased from Solarbio Co., Ltd. (Beijing, China). We purchased 0.2 M potassium phosphate buffer (pH = 7.4) from Aladdin Biochemical Technology Co., Ltd. (Shanghai, China). Xanthine (≥98%) and allopurinol (chromatographically pure) were purchased from Sigma Aldrich Co., Ltd. (St. Louis, MO, USA). Sodium hydroxide, anhydrous ethanol, and boric acid were of analytical grade and purchased from Sinopharm chemical reagent Co., Ltd. (Shanghai, China).

### 2.2. Pretreatment of Raw Materials

The SYCs were thawed overnight at 4 °C and then manually filleted. Next, the filets were boiled to kill enzymes, freeze-dried, and ground into a fine powder (sieved through an 80-mesh sieve). The prepared powder was vacuum sealed and stored at −80 °C before subsequent experiments.

### 2.3. Determination of Raw Materials Protein

The determination of protein was carried out using Kjeldahl nitrogen (Kjeltec 8400 Analyzer Unit, Foss Analytical AB, Hoganas, Sweden) according to the Chinese Standard for Food Safety Determination of Protein in Food (GB 5009.5-2016). Briefly, approximately 500 mg of lyophilized sample was digested by the addition of the digestion mixture and 12 mL of concentrated hydrochloric acid at 420 °C for 80 min and then cooled and subjected to distillation with 50 mL of 40% NaOH and auto-titration experiments using 0.1005 M HCl. 

### 2.4. Preparation of Papain Hydrolysates from SYC 

The SYC peptides were prepared in accordance with the research of Hu et al. [14] with appropriate modifications. The critical hydrolysis parameters for the preparation of papain SYCH were optimized according to our previous unpublished study. The substrate concentration (1:20 *w*/*v*, protein weight basis) was hydrated at 50 °C for 15 min with gentle stirring, adjusted to pH 6.8 with papain at 3000 U/g on a protein basis for 6 h at 50 °C. The mixture was then heated at 95 °C for 10 min to inactivate enzymes and centrifuged at 3950× *g* for 20 min at 4 °C. The supernatant was collected, concentrated, and freeze-dried to obtain SYCHs. SYCHs were stored at −80 °C before subsequent experiments. 

### 2.5. Preparation of Peptide Fractions of SYCHs 

The enzymatic solution was fractionated through an ultrafiltration centrifuge tube with MW cut-offs of 10 kDa, 3 kDa, and 1 kDa (Pall, New York, NY, USA). The fractions corresponding to three MW distributions, i.e., >10 kDa (UF-1), 3–10 kDa (UF-2), and <3 kDa (UF-3), were concentrated and freeze-dried to obtain peptide fractions, which were stored at −80 °C before subsequent experiments.

### 2.6. Determination of Amino Acids Composition of SYC and SYCHs 

Amino acids composition was determined using the method reported by Hou et al. [23] with some modifications, using an Agilent 1100 high-performance liquid chromatography (HPLC) instrument (Wilmington, DE, USA) coupled with a VWD detector (Agilent Technologies, Inc., Wilmington, DE, USA) and a column of Agilent Zoubax Elicpse AAA (4.6 × 150 mm, 3.5 μm). The determination of 17 hydrolysis AAs of 100 mg SYC and SYCHs was performed with 6 M HCl for 22 h, while Trp analysis of 100 mg SYC was performed by alkaline hydrolysis using 5 M NaOH for 20 h. After passing through a 0.22 μm filter, 10 μL of the sample was loaded into the column and eluted at a flow rate of 1.0 mL/min. The temperature was 40 °C, ultraviolet, 338 nm (0–19 min), 266 nm (19.01–25 min); mobile phase A (40 mM sodium dihydrogen phosphate (pH 7.8)); mobile phase B (acetonitrile: methanol: water = 45:45:10). All of the AAs were detected at 338 nm, except Pro, which was detected at 266 nm. The AAs were identified and quantified by authentic AA standards comparing the retention time and peak.

### 2.7. Determination of XOI Activity IC_50_ In Vitro

The XOI activity levels of SYCHs were determined and calculated with the methods reported by Liu and Wei [24,25] with slight modifications. Xanthine was dissolved in 0.2 M potassium phosphate buffer (pH = 7.4) to a concentration of 0.48 mM. In addition, samples (SYCH, UF-1, UF-2, and UF-3) were also dissolved in 0.2 M potassium phosphate buffer (pH = 7.4). Next, 50 μL of sample solution and 50 μL of XO solution (0.07 U/mL) were mixed and incubated at 37 °C for 5 min, then 150 μL of xanthine solution was added to the mixture to continue the reaction. The absorbance of formed uric acid in the samples was monitored at 290 nm with a multifunctional microplate reader (Tecan Co., Ltd., Männedorf, Switzerland). The results were recorded for 10 min. The assay was performed in triplicate. The formula for the calculation of XOI activity is as follows:$$\mathrm{XO}\;50\%\;\mathrm{inhibition}=\frac{{\left(\mathrm{dA}/\mathrm{dt}\right)}_{blank}-{\left(\mathrm{dA}/\mathrm{dt}\right)}_{sample}}{{\left(\mathrm{dA}/\mathrm{dt}\right)}_{blank}}\;\times \;100\%$$
where (dA/dt)*_blank_* and (dA/dt)*_sample_* are the reaction rate without and with the test sample inhibitor, respectively. IC_50_ values were calculated from the mean values of data. XOI activity IC_50_ (the concentration of active compound required to observe 50% XO inhibition) was determined by plotting the percentage inhibition as a function of concentration of the test compound.

### 2.8. Determination of MW Distributions

The MW distributions of SYCH and UF-3, which showed the lowest XOI activity IC_50_ (detailed in Section 3.3), were determined as described by Bao et al. [26] with slight modifications. Gel permeation chromatography (Waters 1515, Waters Co., Milford, MA, USA) with a 2414 differential refractive index detector and an Ultrahydrogel gel permeation chromatography column (7.8 × 300 mm, Waters Co., Milford, MA, USA) was used. The measurement conditions were as follows: 5 mg/mL of the sample (SYCH and UF-3) concentration; mobile phase, 0.1 M sodium nitrate solution; flow rate, 1 mL/min; oven temperature, 40 °C; detector temperature, 40 °C; and the standard, polyethylene glycol.

### 2.9. Identification of the AA Sequence and Molecular Mass of SYCH

UF-3 showed the strongest XOI activity (detailed in Section 3.3). Thus, the AA sequence and molecular mass of UF-3 were identified by a nano-HPLC-MS/MS equipped with a Q Exactive Plus mass spectrometer (Thermo Fisher Scientific, Waltham, MA, USA). The samples were injected into a chromatographic analytical column (C18, 75 µm × 25 cm, 2 μm, 100 Å, Thermo Fisher Scientific) at a flow rate of 300 nL/min. The elution conditions were as follows: mobile phase A (0.1% formic acid in water); mobile phase B (0.1% formic acid in acetonitrile); and a column temperature of 40 °C. The liquid phase separation gradient was as follows: start from 6% to 25% mobile phase B over 42 min, followed by an increase to 45% mobile phase B over 11 min and an increase to 80% mobile phase B over 0.5 min, then hold at 80% mobile phase B for 6.5 min at a sustained flow rate of 300 nL/min. 

Peptides were acquired in data d acquisition (DDA) mode with each scan cycle containing one full MS scan (R = 60 K, AGC (automatic gain control) = 3 × 10^6^, max IT = 20 ms, scan range = 350–1800 mass/charge) and 25 subsequent MS/MS scans (R = 15 K, AGC = 2 × 10^5^, max IT = 50 ms). The mass spectral data were searched by Max Quant (V1.6.6) software. 

### 2.10. Physicochemical Properties Prediction of Active Peptides

Bioinformatics methodologies depend on data maintained in a variety of databases. We conducted computational investigations using database-based search tools; all programs were executed on 30 August 2022. We predicted the hemolytic properties of peptides using an online prediction website (http://codes.bio/hemopred/, accessed on 30 August 2022). We employed the toxicity prediction tool ToxinPred (https://webs.iiitd.edu.in/raghava/toxinpred/index.html, accessed on 30 August 2022) to predict the potential toxicity of XOI peptides [27]. Meanwhile, we predicted the isoelectric point (pI) of the peptides using Prot Param (http://web.expasy.org/protparam/, accessed on 30 August 2022) [12]. We used Innovagen (http://www.innovagen.com/proteomics-tools, accessed on 30 August 2022) to predict the water solubility of the screened potential bioactive peptides [28]. Additionally, we predicted the potential biological activity of all peptides using the Peptide Ranker (http://distilldeep.ucd.ie/PeptideRanker/, accessed on 30 August 2022), with scores between 0 and 1 [12]. The closer the calculated value was to 1, the higher the activity exhibited by the fragments.

### 2.11. Peptides Synthesis 

The two peptides (purity > 95%) Trp-Asp-Asp-Met-Glu-Lys-Ile-Trp (WDDMEKIW, WW8) and Ala-Pro-Pro-Glu-Arg-Lys-Tyr-Ser-Val-Trp (APPERKYSVW, AW10) from the enzymatic hydrolysates of papain SYCHs identified by nano-HPLC-MS/MS (detailed in Section 2.9 and Section 3.4) were chemically synthesized at Sangon Biotech Co., Ltd. (Shanghai, China).

### 2.12. Molecular Docking and Interaction Visual Analysis

We used the docking program Auto Dock Vina to simulate molecular modeling studies in order to further understand the probable binding mechanism of peptides with XO [10]. The X-ray crystal structure of XO from bovine milk with quercetin (PDB: 3 NVY) was downloaded from the RCSB Protein Data Bank (http://www.rcsb.org/pdb) (accessed on 10 September 2022) [28]. The water molecules and all small molecules in XO were removed via Auto Dock tools (v1.5.6). The 3D structures of the inhibitor molecules were built and optimized by minimizing energy in ChemBio3D Ultra 14.0 [11]. Then, the ligands were docked with the XO crystal structure. Peptides and ligand inhibitors were then docked with the PDB structures, giving a Vina score, which is the predicted affinity of the molecule to bind to the PDB structure, calculated in kcal/mol. A more negative score indicates that a ligand is more likely to dock with the enzyme and achieve more favorable interactions [10]. The highest scoring docked model of a ligand was chosen herein to represent its most favorable binding mode predicted by Auto Dock Vina [10]. We carried out functional visualization of the peptides and 3NVY docking results using Pymol2.3.0 [29] to analyze their interaction patterns with binding site residues.

### 2.13. Statistical Analysis

All experimental data were analyzed using SPSS 25.0 (SPSS, Inc., Chicago, IL, USA) and Origin 2021 (Origin Lab, Northampton, MA, USA) software. Data are presented as the mean ± standard deviation (SD). One-way analysis of variance (ANOVA) with least significant difference (LSD) procedures was used to determine the significance of the main effects, and *p* < 0.05 was considered statistically significant.

## 3. Results and Discussion

### 3.1. The Potential of SYC to Decrease Uric Acid Levels

The protein content of SYC was determined to be 88.96% ± 1.40%. Table 1 summarizes the 18 AA composition of SYC. SYC is rich in a variety of AAs, with a total amino acid (TAA) content of 824.45 ± 10.50 mg/g, including 327.39 ± 5.41 mg/g of essential amino acids (EAA) for humans, which accounted for 41.09% of the total. The major AAs of SYC protein were Glu (21.08%), Asp (9.41%), Lys (9.07%), and Leu (8.11%). SYC was rich in HAAs (34.22%), AAAs (9.39%), and BAAs (17.38%), indicating that SYC might be a source of uric-acid-lowering peptides. Hydrophobic amino acids (HAAs) (Met, Leu, and Ala), aromatic amino acids (AAAs) (Trp, Phe, and Tyr), and basic amino acids (BAAs) (Lys, His, and Arg) play essential roles in the uric-acid-lowering process of peptides [23,30,31]. A hydrophobic pocket formed by AA residues near the XO active core acts as a critical structural domain that is accessible to peptides with more HAAs [23]. These AAs can bind to XO via hydrophobic interactions, for example, altering its spatial structure and thereby limiting its activity. Furthermore, it has previously been claimed that proteins with charged AAs and AAAs, particularly Glu, constitute a valuable source of active XOI hydrolysis products [30]. Given that XO generates reactive oxygen species (ROS) by utilizing molecular oxygen as an electron acceptor, the hyperuricemia-treating medicine allopurinol also has antioxidant activity [32]. Because of the significance of phenolic and indole groups as hydrogen donors, AAAs exhibit significant antioxidant action. Furthermore, AAs with charged residues interact with metal ions and restrict oxidative activity [33]. The physiological action of the peptides benefits from a decrease in ROS, which may be related to the effect that decreases uric acid [30]. These results support the evidence indicating that SYC is a potential source of uric-acid-lowering peptides.

### 3.2. The Optimal Conditions for the Preparation of SYCH

Targeted hydrolysis of endogenous proteins employing proteases is a typical approach for generating peptides with specified active bioactivities [34]. As indicated in Figure 1A, papain was more efficient than Neutrase and Alcalase in hydrolyzing muscle proteins, and papain hydrolysates had the lowest IC_50_ values (IC_50_ = 33.40 ± 0.26 mg/mL) for XOI activity, due to variations in the protein sequence in the substrate and the accessibility of the enzyme to the active site. Subsequently, a one-way experiment was conducted on the enzymatic hydrolysis time of papain based on the XOI activity and achieved the best XOI active hydrolysates at 6 h (*p* < 0.05), as shown in Figure 1B, which may be because the SYC muscle was more sufficiently hydrolyzed, and more short peptides with high XOI activity were formed as the hydrolysis time increased. In general, papain degrades proteins more thoroughly than other endoproteases. Additionally, papain has been identified as one of the most appropriate enzymes, with a low cost per unit activity [35]. Moreover, some research has used papain to create XOI peptides. For example, it was demonstrated that protein hydrolysates derived from bonito hydrolysates prepared with papain exhibited XOI activity [36].

The biological activity of peptides in fish hydrolysates primarily depends on their structural properties such as AA content, sequence, and hydrophobicity [37]. Table 2 shows the AA composition of SYCH, which has a total acid hydrolyzable AA content of 782.73 ± 16.20 mg/g, an EAA content of 288.50 ± 8.57 mg/g (36.86%), and an HAA content of 248.92 ± 6.66 (31.81%). Studies confirmed that HAA facilitates the interaction with hydrophobic targets (e.g., cell membranes), thereby enhancing their bioavailability [13]. Additionally, SYCH is high in AAAs and BAAs, which may play a key role in the peptide’s functional capabilities [30,36]. AAAs have benzene rings in their molecules. An AAA at one end of the peptide segment may be more beneficial for binding of the peptide to the enzyme’s active domain since the presence of a benzene ring structure in these peptide segments is thought to have a substantial XO inhibition rate [11]. These findings imply that SYCHs may have bioavailability and anti-hyperuricemia activity, conducive to the next step of obtaining active peptides. 

### 3.3. MW Distribution of XOI Peptides

Low-molecular-weight peptides have been shown to have improved bioactivity and a better capacity to penetrate the gastrointestinal membrane [38]. Bioactive peptides have been refined using UF. Figure 2A shows that the SYCHs mostly consist of peptides of different MWs, with the MW distribution being centered below 3 kDa. MW distributions of SYCHs and UF-3 are depicted in Appendix A, and relative peak table of SYCHs and UF-3 MW distributions in Appendix A, respectively. He et al. [4] suggested that low-molecular-weight peptides may be important contributors to the significant XOI activity of XOI peptides. The peptides in SYCHs were fractionated and the obtained fractions were individually tested in terms of their XOI activities. Figure 2B displays the outcomes of the XOI activity of the SYCH and the three fractions after UF. Comparing these four fractions revealed that the UF-3 (IC_50_ = 25.87 ± 0.16 mg/mL) had the strongest ability to inhibit XO. UF-3 also showed low molecular weight (Figure 2A), which is consistent with the trend of inhibitory activity of XOI. He et al. [4] collected and examined eight fractions with different molecular weight distributions from the lyophilized ethanol-soluble fraction powder of tuna protein hydrolysates and found that the fractions of small peptides (<1 kDa) had remarkable XOI activity compared to the original hydrolysates and other fractions. Other studies found that peptides from skipjack tuna below 3 kDa had a greater inhibitory effect on XO than other fractions [34]. Similar to previous findings, our investigation found that the fraction with 3 kDa inhibited XO more than the SYCH and other fractions, indicating that the fraction with 3 kDa has structural properties recognized by XO and thus functions as a substrate for XO. Therefore, we selected UF-3 for the next identification stage.

The AA compositions of SYCH and UF-3 (showing the strongest XOI activity) were evaluated by the acid hydrolysis method to confirm whether there is a potential link between AA composition and uric acid decrease. The makeup of the AA group shifted with the change in MW, Glu, Asp, and Lys were the most prevalent AAs in both SYCH and UF-3, as indicated in Table 2. Following UF, the proportions of BAA, AAA, and HAA increased by 2.64%, 5.72%, and 6.48%, respectively, in UF-3 (Table 2). A molecular docking approach was used to mimic the structure–activity relationships of 20 amino acids, 400 dipeptides, and 8000 tripeptides with XO [9,10]. AAA and HAA were shown to be more likely to connect with the critical amino acid residues around the active core of XO, resulting in a substantial inhibitory action against XO. This could be the reason for the higher XOI activity of UF-3 than others.

### 3.4. Identification of XOI Peptides Sequences and Validation of XOI Activity

Seven peptides (WDDMEKIW, APPERKYSVW, IADRMQKELT, LNSADLIK, LSNLGIVI, IGALRAVA, and HHTFYNELR) derived from SYC were obtained and their activity values, hemolysis, and toxicity were predicted. Their basic data (AA sequence, MW, PI, predicted activity values, hemolysis, toxicity, and IC_50_ for XOI activity) are summarized in Table 3. According to Table 3, all of the peptides had a molecular weight between 760 and 1250 and were predicted to be non-toxic. Only WW8 and AW10 were predicted to have higher potential physiological activity, and all were non-hemolytic except LSNLGIVI and LNSADLIK. The results indicated that WW8 and AW10 (the identification results in Figure 3) are non-toxic and non-hemolytic and have potential physiological activity. WW8 and AW10 presented adequate water solubility. Water solubility has been found to have an effect on bioactive peptide absorption, and dissolution is a limiting factor for physiological function performance [39]. Peptides with considerable water solubility may have high biological availability [15,40]. The XOI activities of the peptides were highest in WW8 (IC_50_ = 3.16 ± 0.03 mM) and AW10 (IC_50_ = 5.86 ± 0.02 mM), as shown in Figure 4. The results showed a higher XOI effect than peptide ACECD from Skipjack tuna hydrolysates, which had XOI activity of IC_50_ of 13.40 mM [13]. IADRMQKELT and LNSADLIK had no XOI activity at 15 mg/mL, and LSNLGIVI, IGALRAVA, and HHTFYNELR did not even dissolve in water at 3 mg/mL, so these five peptides were neglected for the subsequent experiments.

The XOI activity of WW8 and AW10 was significantly higher than that of the other peptides lacking Trp residue. These findings suggest that a crucial component of effective XOI activity is the presence of Trp residues in the peptides [9,10]. Nongonierma [9] reported that Trp inhibited XO by 70.3 ± 1.1% at a concentration of 0.25 mg/mL. Li [10] claimed that relatively lower IC_50_ values were mainly located in the peptides containing Trp residue, reporting that the IC_50_ values of peptides WDD, WDQW, PPKNW, WPPKN, and WSREEQE were lower than those of peptides HCPF and ADIYTE, and that these Trp-containing peptides had relatively higher XOI activity. One possible explanation for this is that Trp with an indole group has a similar C6 and C5 ring structure to the drug allopurinol. Hou et al. [23] came to the same conclusion and demonstrated that peptides with Trp residue at the C-terminus inhibited XO. Similarly, the WW8 and AW10 had Trp residue at the C-terminus, which was the most critical factor contributing to the inhibitory effect. Furthermore, prior research suggested that peptides with HAA function well as XO inhibitors, because peptides with higher amounts of HAA may be able to access the hydrophobic domain of the XO active center more easily [36,41]. Interestingly, the peptides with high XOI activity all contain these specific residues. WW8 and AW10 were abundant in HAA (50% HAA residues), including Try, Tyr, Pro, and Ile, which is consistent with prior results on XOI peptides. 

The higher suppression of XO peptides compared to SYCH implies that peptides generated from SYC, notably WW8 and AW10, could be potential XO inhibitors. Thus, the peptides WW8 and AW10, which are non-toxic and non-hemolytic and have adequate good water solubility and XOI activity, were subjected to molecular docking to explain the relationship between peptides and XO. 

### 3.5. Molecular Docking and Visual Analysis

Molecular docking simulates and visualizes the binding sites and binding profiles of small molecule ligands to biological macromolecule receptors. Figure 5A,B depicts the interaction between the two peptides (WW8 and AW10) with XO, with binding energies of −7.3 and −7.9 kcal/mol (detailed in Table 4), respectively, indicating a strong binding relationship. Generally, the lower the binding energy for the same docking model, the more stable the complex [23].

The forces created between the peptides and XO (PDB: 3NVY), as well as the interactions and matching bonds, are depicted in Figure 5C and Table 4. The peptides WW8 and AW10 interacted with the protein, and WW8 established hydrogen connections with Ile1190, Ala1189, Leu744, and Gln1201, with hydrogen bond lengths of 3.5 Å, 2.7 Å, 2.0 Å, and 2.4 Å, respectively. WW8 also established electrostatic interaction with His579 and hydrophobic interactions with 24 AAs, including Val1200, Gly1197, Glu1196, Phe1219, Ile1235, Ile1229, Phe1232, Pro1230, His741, Ala1231, Tyr743, Phe238, Phe742, Tyr592, Met1038, Gly1039, Gly796, Gly1039, Met794, Gly795, Ala582, Gln585, Gln1194, and Gly1193. These AA residues appeared around the binding of AW10 to XO, including Arg912 and Met1038 with which they formed hydrogen bonds with lengths of 1.9 Å, 2.1 Å, and 3.4 Å, respectively. Moreover, hydrophobic interactions with 22 AAs were distributed around the binding sites of AW10 in XOD, including Ala582, His579, Gln585, Met794, Gly796, Gly795, Leu744, Tyr743, Tyr592, Gly039, Gln194, Gly193, Gln021, Phe798, Ala1198, Glu1196, Ile1235, Phe1239, Gly1197, Val1200, Phe1232, and Ala1231, indicating that the hydrophobic force is another important factor driving the binding of AW10 to XO.

It was hypothesized that although all peptides interact with different AA residues of XO, they all bind to XO mainly through hydrogen bonding and hydrophobic forces, thus inhibiting the catalytic activity of XO. The lower XOI activity of IC_50_ for WW8 compared to AW10 could be attributed to a greater number of hydrogen bonds and hydrophobic forces between the WW8 and the XO interaction than AW10.

## 4. Conclusions

Peptides (WDDMEKIW and APPERKYSVW) with XOI activity were identified after enzymatic hydrolysis and UF separation of SYC proteins, and the IC_50_ values (IC_50_ = 3.16 ± 0.03 mM and 5.86 ± 0.02 mM, respectively) of XOI activity were calculated in vitro. These findings were validated by molecular docking of the two peptides chosen for the strongest XOI activity, which highlighted the importance of hydrophobic bonds and hydrogen bonds in the establishment of a stable complex conformation and the resulting inhibitory effect of the peptides. We anticipate that these peptides can be employed to manage hyperuricemia as natural XO inhibitors.

Bioactive peptides will continue to constitute an important area of study in the future, with an expanding array of uses in food, medicine, and cosmetics. Although peptides derived from SYC proteins exhibit XOI activity, in vivo experiments and clinical trial data are required to confirm these findings and explain unknown mechanisms. Clinical trial data are also necessary to demonstrate the efficacy of active peptides and ensure their bioavailability and safety profile. These are all objectives that must be fulfilled in the future. In terms of preserving peptide activity, micro- and nano-encapsulation of bioactive peptides may be an effective way to manage their release and avoid degradation in order to optimize their bioavailability and effectiveness. The advancement of oral administration and bioactive peptide delivery introduces both possibilities and limitations to be addressed in future research.

## Figures and Tables

**Figure 1 foods-12-00982-f001:**
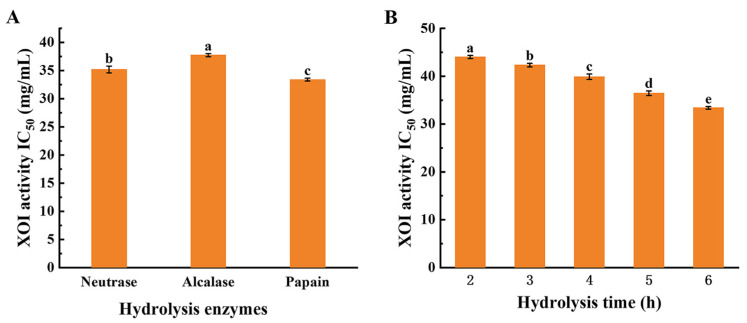
The inhibitory activity against xanthine oxidase with various hydrolysis enzymes and hydrolysis times of small yellow croaker hydrolysates (SYCHs). (**A**) Xanthine oxidase inhibitory (XOI) activity IC_50_ of various hydrolysis enzymes in SYCHs for 6 h. (**B**) XOI activity IC_50_ of papain SYCH. Data are shown as the mean ± standard deviation (*n* = 3), and different letters (a–e) among samples indicate significant differences (*p* < 0.05).

**Figure 2 foods-12-00982-f002:**
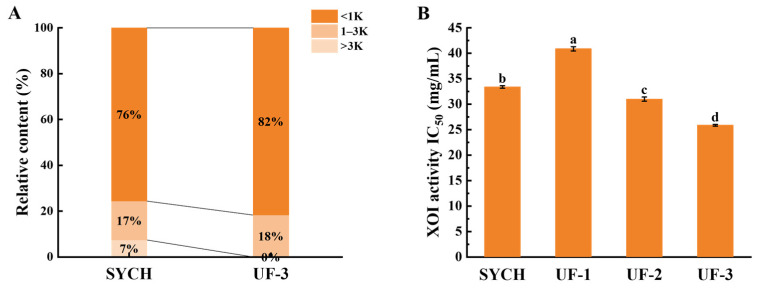
Molecular weight (MW) and XOI activity IC_50_ of papain hydrolysates from small yellow croakers at 6 h. (**A**) MW distributions and relative contents of different samples. (**B**) XOI activity IC_50_ of SYCH and three fractions after ultrafiltration. Different letters (a–d) among samples indicate significant differences (*p* < 0.05).

**Figure 3 foods-12-00982-f003:**
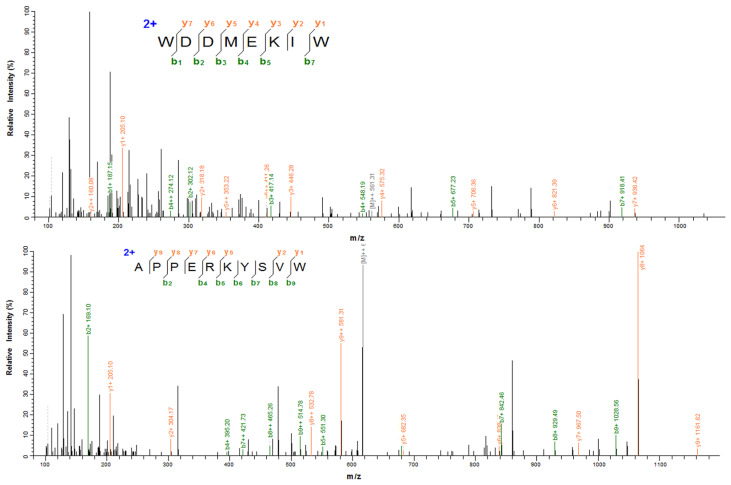
Identification results of WDDMEKIW and APPERKYSVW.

**Figure 4 foods-12-00982-f004:**
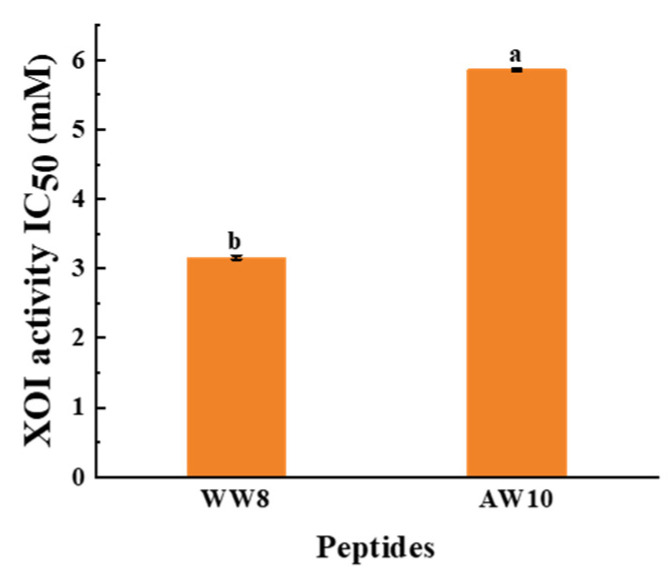
XOI activity IC_50_ of two synthesized XOI peptides. Data are shown as the mean ± standard deviation (*n* = 3), and different letters (a and b) among the samples indicate significant differences (*p* < 0.05).

**Figure 5 foods-12-00982-f005:**
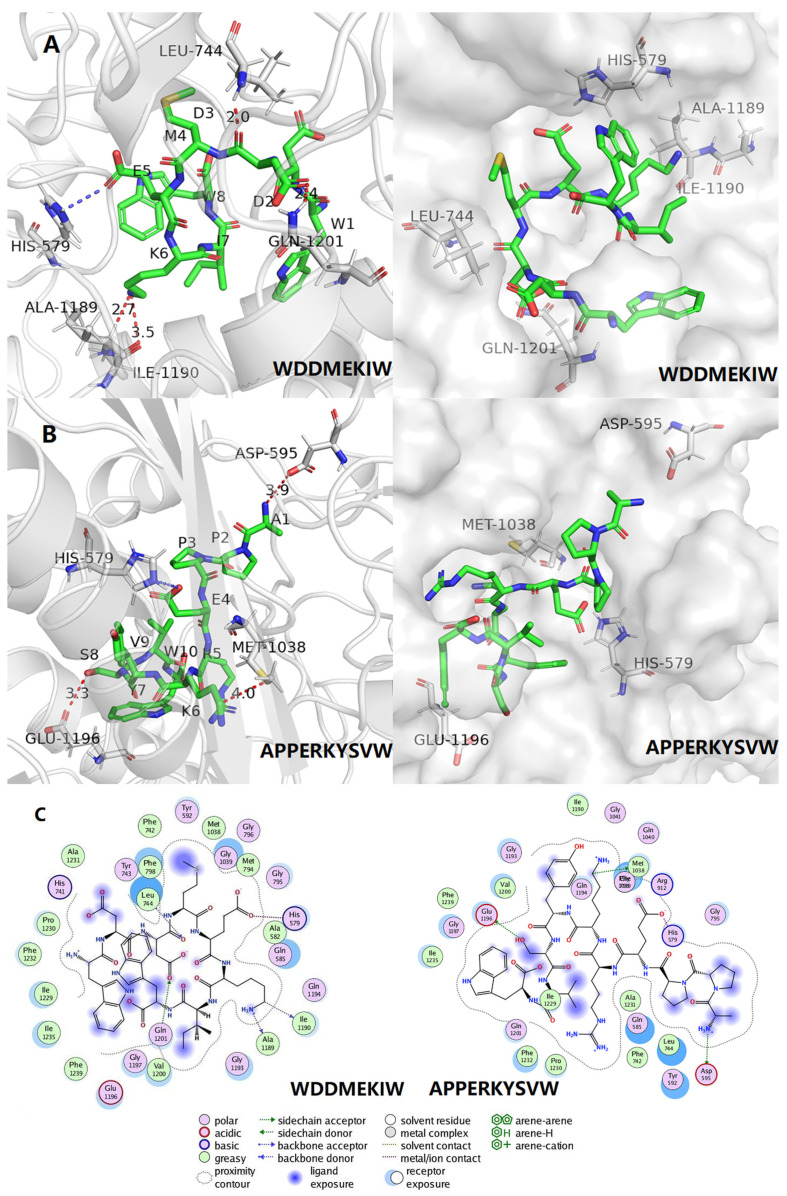
The molecular docking analysis of the synthesized peptides with xanthine oxidase (XO). (**A**) Molecular docking of the peptide WDDMEKIW with XO. (**B**) Molecular docking of the peptide APPERKYSVW with XO. (**C**) The binding and interaction visualization of peptides with XO.

**Table 1 foods-12-00982-t001:** Amino acid composition of small yellow croakers.

AA	Content (mg/g)	Proportion	AA	Content (mg/g)	Proportion
Glu	173.82 ± 1.00	21.08%	Val	37.03 ± 0.25	4.49%
Asp	77.60 ± 0.66	9.41%	Phe	35.60 ± 0.83	4.32%
Lys	74.77 ± 0.51	9.07%	Ser	35.68 ± 0.25	4.33%
Leu	66.83 ± 0.75	8.11%	Tyr	31.02 ± 0.47	3.76%
Ala	52.98 ± 0.43	6.43%	Met	28.06 ± 0.55	3.40%
Arg	52.85 ± 0.53	6.41%	His	15.65 ± 0.25	1.90%
Thr	37.78 ± 0.99	4.58%	Pro	14.31 ± 0.43	1.74%
Gly	38.05 ± 0.75	4.62%	Cys	5.08 ± 0.13	0.62%
Ile	36.53 ± 0.87	4.43%	Trp	10.78 ± 0.86	1.31%
EAA	327.39 ± 5.41	39.71%	HAA	282.14 ± 4.85	34.22%
BAA	143.27 ± 1.24	17.38%	AAA	77.40 ± 2.10	9.39%
TAA	824.45 ± 10.50	100%			

Data are shown as the mean ± standard deviation (*n* = 3). AA: amino acid. EAA (essential amino acid): Lys; Trp; Phe; Met; Ile; Thr; Val; Leu. HAA: (hydrophobic amino acid): Ala; Ile; Leu; Met; Phe; Val; Trp; Pro. AAA (aromatic amino acids): Phe; Trp; Tyr. BAA (basic amino acid): Lys; Arg; His. TAA: total amino acid.

**Table 2 foods-12-00982-t002:** Amino acid composition of small yellow croaker hydrolysates and UF-3.

Samples	SYCH		UF-3		Samples	SYCH		UF-3	
AA	Content (mg/g)	Proportion	Content (mg/g)	Proportion	AA	Content (mg/g)	Proportion	Content (mg/g)	Proportion
Glu	164.78 ± 1.77	21.05%	127.27 ± 3.02	18.87%	Ile	26.50 ± 1.23	3.39%	24.01 ± 0.40	3.56%
Lys	79.46 ± 1.57	10.15%	68.56 ± 0.65	10.17%	Tyr	24.80 ± 0.31	3.17%	22.98 ± 0.24	3.41%
Asp	80.55 ± 2.18	10.29%	63.64 ± 0.89	9.44%	His	17.77 ± 0.97	2.27%	17.25 ± 0.32	2.56%
Leu	63.08 ± 0.89	8.06%	54.77 ± 0.16	8.12%	Met	17.63 ± 1.03	2.25%	14.66 ± 0.60	2.17%
Ala	57.23 ± 1.03	7.31%	55.45 ± 1.26	8.22%	Pro	19.77 ± 0.91	2.53%	15.92 ± 0.34	2.36%
Arg	50.84 ± 0.76	6.49%	45.18 ± 0.40	6.70%	Cys	5.28 ± 0.53	0.67%	5.45 ± 0.28	0.81%
Thr	37.12 ± 1.20	4.74%	31.67 ± 0.17	4.70%	EAA	288.50 ± 8.57	36.86%	255.81 ± 1.13	37.84%
Gly	36.86 ± 0.30	4.71%	33.86 ± 0.94	5.02%	HAA	248.92 ± 6.66	31.80%	228.32 ± 2.24	33.86%
Ser	36.36 ± 0.68	4.64%	30.17 ± 0.11	4.47%	AAA	54.67 ± 1.67	6.99%	49.81 ± 0.02	7.39%
Val	34.83 ± 1.45	4.45%	36.67 ± 0.99	5.44%	BAA	148.07 ± 3.00	18.92%	130.98 ± 1.32	19.42%
Phe	29.88 ± 1.41	3.82%	26.83 ± 0.23	3.98%	TAA	782.73 ± 16.20	100%	676.00 ± 2.96	100%

Data are shown as the mean ± standard deviation (*n* = 3). SYCH: small yellow croaker hydrolysates. UF-3: peptides of a molecular weight less than 3 kDa after ultrafiltration of SYCH. AA: amino acid. EAA (essential amino acid): Lys; Phe; Met; Ile; Thr; Val; Leu. HAA: (hydrophobic amino acid): Ala; Ile; Leu; Met; Phe; Val; Pro. AAA (aromatic amino acids): Phe; Tyr. BAA (basic amino acid): Lys; Arg; His. TAA: total amino acid.

**Table 3 foods-12-00982-t003:** Prediction of physicochemical properties of active peptides.

Serial Number	Amino Acid Sequence	Amino Acid Number	Molecular Weight	Isoelectric Point	Predicted Activity Values	Hemolysis	Toxicity	XOI Activity of IC_50_ (mM)
1	WDDMEKIW	8	1122.25	4.03	0.73	Non-hemolytic	Non-toxic	3.16 ± 0.03
2	APPERKYSVW	10	1232.40	8.63	0.58	Non-hemolytic	Non-toxic	5.86 ± 0.02
3	IADRMQKELT	10	1204.41	6.07	0.15	Non-hemolytic	Non-toxic	NA
4	LNSADLIK	8	873.02	5.84	0.19	Non-hemolytic	Non-toxic	NA
5	LSNLGIVI	8	828.02	5.52	0.18	Hemolytic	Non-toxic	NA
6	IGALRAVA	8	769.94	9.75	0.14	Hemolytic	Non-toxic	NA
7	HHTFYNELR	9	1216.3	6.92	0.38	Non-hemolytic	Non-toxic	NA

NA: not applicable.

**Table 4 foods-12-00982-t004:** The interaction types and amino acid residues of peptides with XO.

Peptides	Binding Energy (kcal/mol)	Hydrogen Bonding	Hydrophobic Interactions	Electrostatic Interaction
WDDMEKIW	−7.3	Ile1190, Ala1189, Leu744, Gln1201	Val1200, Gly1197, Glu1196, Phe1219, Ile1235, Ile1229, Phe1232, Pro1230, His741, Ala1231, Tyr743, Phe238, Phe742, Tyr592, Met1038, Gly1039, Gly796, Gly1039, Met794, Gly795, Ala582, Gln585, Gln1194, Gly1193	His579
APPERKYSVW	−7.9	Arg912, Met1038	Ala582, His579, Gln585, Met794, Gly796, Gly795, Leu744, Tyr743, Tyr592, Gly039, Gln194, Gly193, Gln021, Phe798, Ala1198, Glu1196, Ile1235, Phe1239, Gly1197, Val1200, Phe1232, Ala1231	NP

NP: not present.

## Data Availability

All data generated or analyzed during this study are included in this published article.

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
