# Peer review of "Xanthine Oxidase Inhibitory Peptides from Larimichthys polyactis: Characterization and In Vitro/In Silico Evidence"

_foods, 2023, doi:10.3390/foods12050982_

Round 1

Reviewer 1 Report

The manuscript describes novel peptide with xanthine oxidase inhibition activity were identified after hydrolysis using papain and separation of SYC protein. the authors present a number of experiments, which can be of some use. However, I think there are some information that need to be clarified. Here are my comments:

1. Proteins can be denatured or degraded by heat. Are there any changes that occur during the boiling process in the SYC pretreatment? ( 2.2. Pretreatment of raw materials). What are the specific boiling conditions? Also, is there a result that when maintained at 95 degrees for 10 minutes after hydrolysis of papain, only papain activity is selectively lost and enzymatic reactants are not affected? ( 2.4. Preparation of papain hydrolysates from SYC)

 2. Figure 2 shows the MW distributions and relative contents of different samples as final converted values. Please add the gel permeation chromatograph chart together.

 3. It is necessary to explain whether the standard of GPC is PEG, not peptide.

Author Response

Response to Reviewer 1 Comments

Dear reviewer,

Thanks for your careful review of our manuscript. We found that all your critical comments and suggestions were constructive. Accordingly, we have modified the manuscript to the current version, and the detailed corrections have been listed point by point. All the changing parts in the revised manuscript were marked in red color. We sincerely hope this version of the manuscript will be satisfactory.

Response to Reviewer

The manuscript describes novel peptide with xanthine oxidase inhibition activities were identified after hydrolysis using papain and separation of SYC protein. the authors present a number of experiments, which can be of some use. However, I think there are some information that need to be clarified. Here are my comments:

Thank you very much for your valuable considerations and comments. We have revised our manuscript accordingly. Regarding other specific questions, our answers and changes are listed as follows:

  1. Proteins can be denatured or degraded by heat. Are there any changes that occur during the boiling process in the SYC pretreatment? ( 2.2. Pretreatment of raw materials). What are the specific boiling conditions? Also, is there a result that when maintained at 95 degrees for 10 minutes after hydrolysis of papain, only papain activity is selectively lost and enzymatic reactants are not affected? ( 2.4. Preparation of papain hydrolysates from SYC)

Response: (1) Changes that occurred during the boiling process in the SYC pretreatment: The flesh gradually changed from raw to cooked, the fish hardened and turned white, the meat was soft and tender, and the bones separated easily, while a distinct fresh fish flavor emerged. (2) The specific boiling conditions: Heat the small yellow croaker in a moderate amount of cold water to boiling for 5 minutes at 100 °C. (3) In the process of inactivating the enzyme, enzymatic reactants were not affected. Because the small yellow croaker hydrolysates were peptides of varied length that were thermally stable and retained their primary structure after 10 minutes of heating at 95 °C.

  1. Figure 2 shows the MW distributions and relative contents of different samples as final converted values. Please add the gel permeation chromatograph chart together.

Response: The gel permeation chromatograph chart has been added to the supplementary material.

  1. It is necessary to explain whether the standard of GPC is PEG, not peptide.

Response: Thank you for your kind suggestion. The standard of GPC is PEG.

Reviewer 2 Report

It seems to me a fascinating document,

The work aimed to characterize Papain small yellow croaker hydrolysates (SYCH), finding that they have potent xanthine oxidase inhibitory (XOI).

It is necessary that the abstract clearly describes the objective of the work.

The introduction seems long enough to me. However, the author should extensively review the research background on these peptides.

Please add the origin of all reagents (brand) in the materials and methods section.

In the results section, complete the description of table 1, table 3, table 4, and figure 3; remember that figures and tables should describe themselves, even without consulting the text.

I suggest the author extend the conclusion, including the future challenges of these compounds and offering the possible applications of the extracts.

Author Response

Response to Reviewer 2 Comments

Dear reviewer,

Thanks for your careful review of our manuscript. We found that all your critical comments and suggestions were constructive. Accordingly, we have modified the manuscript to the current version, and the detailed corrections have been listed point by point. All the changing parts in the revised manuscript were marked in red color. We sincerely hope this version of the manuscript will be satisfactory.

Response to Reviewer

It seems to me a fascinating document,

The work aimed to characterize Papain small yellow croaker hydrolysates (SYCH), finding that they have potent xanthine oxidase inhibitory (XOI).

It is necessary that the abstract clearly describes the objective of the work.

Response: Thank you for your kind suggestion. We have added the purpose of the abstract (Line 13-16).

The introduction seems long enough to me. However, the author should extensively review the research background on these peptides.

Response: We have appropriately streamlined the introduction and carefully revised the research background of the peptides (Line 39-47).

Please add the origin of all reagents (brand) in the materials and methods section.

Response: We have supplemented the origin of all reagents (brand) in Line 82-85.

In the results section, complete the description of table 1, table 3, table 4, and figure 3; remember that figures and tables should describe themselves, even without consulting the text.

Response: We completed and marked bright the description of table 1 in Line 215-220, table 3 in Line 327-332, table 4 in Line 377 and 380-394, and figure 3 in Line 330-331.

I suggest the author extend the conclusion, including the future challenges of these compounds and offering the possible applications of the extracts.

Response: We sincerely appreciate the valuable comments. Following your suggestions, we have extended the conclusion (Line 414-426).

Reviewer 3 Report

The aim of this study was to Chemically characterize (AA composition, molecular mass, sequence) the xantine oxidase (XO;  EC 1.17.3.2) inhibitory activity (IC50) of papain-hydrolyzed peptide fractions [>10 kDa (UF-1), 3-10 kDa (UF-2), and < 3 kDa (UF-3)] from small yellow croaker (Larimichthys Polyactis) muscle hydrolysates (SYCH) and to explore the most plausible inhibitory mechanism by mean of in silico molecular docking studies. The study was well designed, and the results/discussion are quite detailed, yet minor changes could improve the study´s scientific soundness and uniqueness:

General. A) The reading and comprehension of the manuscript will improve if it is independently reviewed by a formal translation agency or by a native English-speaking colleague, B) Recheck the manuscript for typos, C) Reduce as much as possible unneeded abbreviations throughout the manuscript (e.g. UA, HUA).

Title. Suggestion: Xantine oxidase inhibitory peptides from Larimichthys Polyactis: Characterization and in vitro/in silico evidence.

Abstract. A) Reduce unneeded abbreviations (e.g.  nano-HPLC-MS/MS).

Introduction & methods. OK.

Results & Discussion. Improve the discussion in QSAR terms to understand the differences between peptide fractions.

Tables and figures: A) The images must be of better resolution (300 dpi), particularly the HPLC-MS2 chromatograms, B) Format the tables according to this journal.

References. It is recommended to review again as some do not have an appropriate format for Foods.

Author Response

Response to Reviewer 3 Comments

Dear reviewer,

Thanks for your careful review of our manuscript. We found that all your critical comments and suggestions were constructive. Accordingly, we have modified the manuscript to the current version, and the detailed corrections have been listed point by point. All the changing parts in the revised manuscript were marked in red color. We sincerely hope this version of the manuscript will be satisfactory.

Response to Reviewer

The aim of this study was to Chemically characterize (AA composition, molecular mass, sequence) the xanthine oxidase (XO; EC 1.17.3.2) inhibitory activity (IC50) of papain-hydrolyzed peptide fractions [>10 kDa (UF-1), 3-10 kDa (UF-2), and < 3 kDa (UF-3)] from small yellow croaker (Larimichthys Polyactis) muscle hydrolysates (SYCH) and to explore the most plausible inhibitory mechanism by mean of in silico molecular docking studies. The study was well designed, and the results/discussion are quite detailed, yet minor changes could improve the study´s scientific soundness and uniqueness:

Thank you very much for your valuable considerations and comments. We have revised our manuscript accordingly. Regarding other specific questions, our answers and changes are listed as follows:

General. A) The reading and comprehension of the manuscript will improve if it is independently reviewed by a formal translation agency or by a native English-speaking colleague, B) Recheck the manuscript for typos, C) Reduce as much as possible unneeded abbreviations throughout the manuscript (e.g. UA, HUA).

Response: We sincerely appreciate the valuable comments. A) We have tried our best to polish the language and reviewed by a formal service in the revised manuscript. B) We have carefully rechecked the manuscript for typos. C) we have reduced as much as possible unneeded abbreviations throughout the manuscript, including UA, HUA.

Title. Suggestion: Xanthine oxidase inhibitory peptides from Larimichthys Polyactis: Characterization and in vitro/in silico evidence.

Response: Thanks, we think this is an excellent suggestion. We have changed the title (Line 2-3).

Abstract. A) Reduce unneeded abbreviations (e.g. nano-HPLC-MS/MS).

Response: We deleted unneeded abbreviations from the abstract, including nano-HPLC-MS/MS, HAA, AA and HUA.

Introduction & methods. OK.

Response: Thank you very much for your recommendation.

Results & Discussion. Improve the discussion in QSAR terms to understand the differences between peptide fractions.

Response: We have improved the discussion in QSAR terms (Line 397-399).

Tables and figures: A) The images must be of better resolution (300 dpi), particularly the HPLC-MS2 chromatograms, B) Format the tables according to this journal.

Response: We sincerely thank the reviewer for careful reading. As suggested by the reviewer, we have replaced images with better resolution and formatted the tables according to this journal.

References. It is recommended to review again as some do not have an appropriate format for Foods.

Response: We have checked and corrected the references carefully in the revised manuscript.

Round 2

Reviewer 1 Report

The revised manuscript is generally improved compared with a previous manuscript.